# Towards Viable Eco-Friendly Local Treatment of Blackwater in Sparsely Populated Regions

Prasaanth Ravi Anusuyadevi [1,*], Darsi Jaya Prasanna Kumar [2,*], A. D. H. V. Omkaara Jyothi [2], Neha S. Patwardhan [2], Janani V. [2] and Arjan Mol [1]

[1] Materials Science and Engineering Department (MSE), Faculty of Mechanical, Maritime and Materials Engineering (3mE), Delft University of Technology, 2628 CD Delft, The Netherlands

[2] Department of Chemical Engineering, M.S. Ramaiah Institute of Technology, Bengaluru 560054, India

* Correspondence: p.ravianusuyadevi@tudelft.nl or ranu.prasaanth@gmail.com (P.R.A.); jp@msrit.edu or djpkumar@gmail.com (D.J.P.K.)

**Abstract:** The maintenance of people's lifestyle against global climate change, exhaustion of groundwater, depletion of minerals, and water scarcity has instigated the recycling and reuse of water from unlikely sources. This situation has motivated researchers to develop effective technologies for treating wastewater, enabling its reuse. Water security has been ensured in myriad, highly populated regions through large-scale centralized treatment facilities. The development and implementation of small-scale, renewable-energy-based, decentralized, on-site treatment methodologies ensure water sustainability in rural areas, where centralized treatment facilities are impractical for application. This review article focuses on the recently reported low-cost purification techniques for recycling wastewater generated by single and community-based households in sparsely populated areas. Here we propose treatment technologies for efficient waste management that can be easily integrated in the upcoming years to the lavatories built under the Swachh Bharat Mission (SBM), a momentous cleanliness campaign that has been successfully implemented by the Government of India (GOI). Specifically, we suggest an electrochemical (EC) method to treat the supernatant of the Blackwater (BW) to produce purified non-potable water for reuse in diverse purposes. The EC technique does not require external chemicals for treatment and can be powered by sustainable technologies (like solar panels), thus reducing the treatment cost. Subsequently, vermicomposting, microwave, biogas, and phycoremediation methods are considered to treat the solid sludge to produce value-added products such as enriched organic fertilizer for agriculture and biofuel. The above methods also ensure the satisfactory reduction in Biochemical Oxygen Demand (BOD) (>85%) and Chemical Oxygen Demand (COD) (81–91%) and the complete removal of pathogens and other harmful pollutants. Finally, the novel treatment techniques discussed here are not only limited to rural areas of India but can be implemented in any rural area of the world.

**Keywords:** Blackwater (BW); Swachh Bharat Mission (SBM); electrochemical (EC); vermicomposting; microwave; biogas; phycoremediation; sustainable water treatment; wastewater management

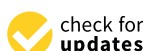



## 1. Introduction

Water is the main constituent of earth's hydrosphere and is essential for all life on earth. All living organisms are made up of water. It is necessary for all chemical reactions that occur in living cells and for food production and other essential activities for the sustenance of life. Humans use water for drinking, washing, cooking, bathing, and flushing the toilet. The wastewater generated due to everyday life-sustaining human activities are classified into two types: 1. Greywater (GW) and 2. Blackwater (BW) [1]. The former corresponds to the wastewater generated from household kitchen sinks, bathtubs and showers, automatic washers, and manual cloth washes. The treatment processes of GW are facile in nature due to its lower degree of contamination. BW refers to the wastewater generated from lavatories,

and it is mainly composed of human excretory products, flush water, and toilet tissues (on some continents). BW contains airborne bacteria, pathogens, and other minerals which, upon release to ponds and rivers, would pollute the water bodies and affect the aquatic life. Consumption of such polluted water with elevated quantities of minerals like nitrates ($NO_3$) carries the potential of carcinogenic threats and blue baby syndrome [2,3]. Humans exposed by other means to BW can suffer from infectious diseases, illness, and long-term respiratory problems. It accounts for a maximum of 30% of domestic wastewater [4]. The rapid increase in the world population coupled with intensive industrialization across the world in the recent decades has led to insufficient availability of water. In many parts of the world, merely due to inadequate water management, new strategies to reuse wastewater have been developed in recent years [5–8]. Studies on sustainable wastewater treatment processes involving natural and man-made waste stabilization ponds with baffles have been carried out intensively in recent decades [9,10]. Be it GW or BW, it must be purified before its reuse. Owing to fewer pollutants in GW, it is recycled and reused for non-potable water applications, such as lavatory usage and agricultural purposes [6,11]. The BW, on the other hand, has been collected by the centralized sewer systems and subsequently treated at sewage treatment plants located in various parts of the urban areas. Finally, it is released into the rainwater drain system that recirculates the water into natural water bodies like lakes, rivers, seas, etc. However, the above centralized treatment of wastewater (especially BW) and sophisticated treatment of BW for its sustainable reuse are lacking in rural areas of the world due to the inherently dispersed population. Huge investments were needed for laying sewer lines across a vast but sparsely populated area, for example in Western Europe, in which nowadays only a small fraction of private citizens purify their wastewater themselves, as they live far away from the country's district water boards and sewage treatment plants [12]. The World Bank estimates, based on the United Nations Population Division's world urbanization prospects, that the current 2022 rural population of the world is 3.4 billion. India has the highest rural population in the world, accounting for more than 900 million people [13].

There has been a steady-state linear increase in the total population in India, which is reflected in a more than two-fold rise in the number of rural residents in the last six decades (see Figure 1, left axis). India being the world's fifth biggest economy [14], its aggressive industrialization and development of diverse manufacturing units have led to rapid urbanization due to which there has been a consistent demographic transition of the rural population to urban centers. This can be visualized in the reduction in the percentage of the total population living in rural areas, as shown in Figure 1 on the right axis. However, the majority of people in India are still rooted in its rural locations; in 2021, 64.65% of its people consisted of masses living in villages [13,15]. Their lifestyle is simplistic, and they are highly dependent on the environment of their locality, as their occupation is primarily agriculture, weaving, handicrafts, small-scale manufacturing units (for example, brick and tile manufacturing), etc. In order to render the modern lifestyle accessible to the rural masses and provide them with a healthier living culture, on October 02, 2014, the Government of India (GOI) launched **Swachh Bharat Mission** (SBM), a **Clean India Mission** aimed at creating excellent sanitation facilities with high hygiene and an open-defecation-free (ODF) India by 2 October 2019 [16]. Through this program, GOI has constructed 109 million individual household lavatories all across the country, resulting in providing access to latrines in all rural households. The distribution of the number of lavatories built in India across all states is depicted in Figure 2.

These lavatories are designed and built in such a way that they have a provision for water storage, especially for cleaning the toilet and for hand washing. Due to such a mass program by GOI, all villages were declared ODF by October 2019. The Department of Drinking Water and Sanitation of GOI with the help of an independent verification agency supported by the World Bank organized three rounds of a National Annual Rural Sanitation Survey (NARSS) within the period of 2017–2022. The survey results of 2019–2020 clearly indicated that 99.6% of households had access to lavatories with water availability facilities.

Subsequently, GOI launched the Jal Jeevan Mission with the ultimate aim of providing household tap water connection for rural households by 2024 [16]. As mentioned in the previous paragraphs, with rapid growth in population and climate change, the available water resources are becoming polluted with domestic, agricultural, and industrial effluents, in turn limiting the availability of usable water resources. Handling the water scarcity issues is the biggest challenge for the rural population, especially in the places where living is dependent on rain-fed agriculture. For instance, the rural population of India are highly dispersed in their locality; most of the villages have a population hardly exceeding 1000 people [17]. The distribution of vast numbers of villages and urban areas in India is shown in Figure 3.

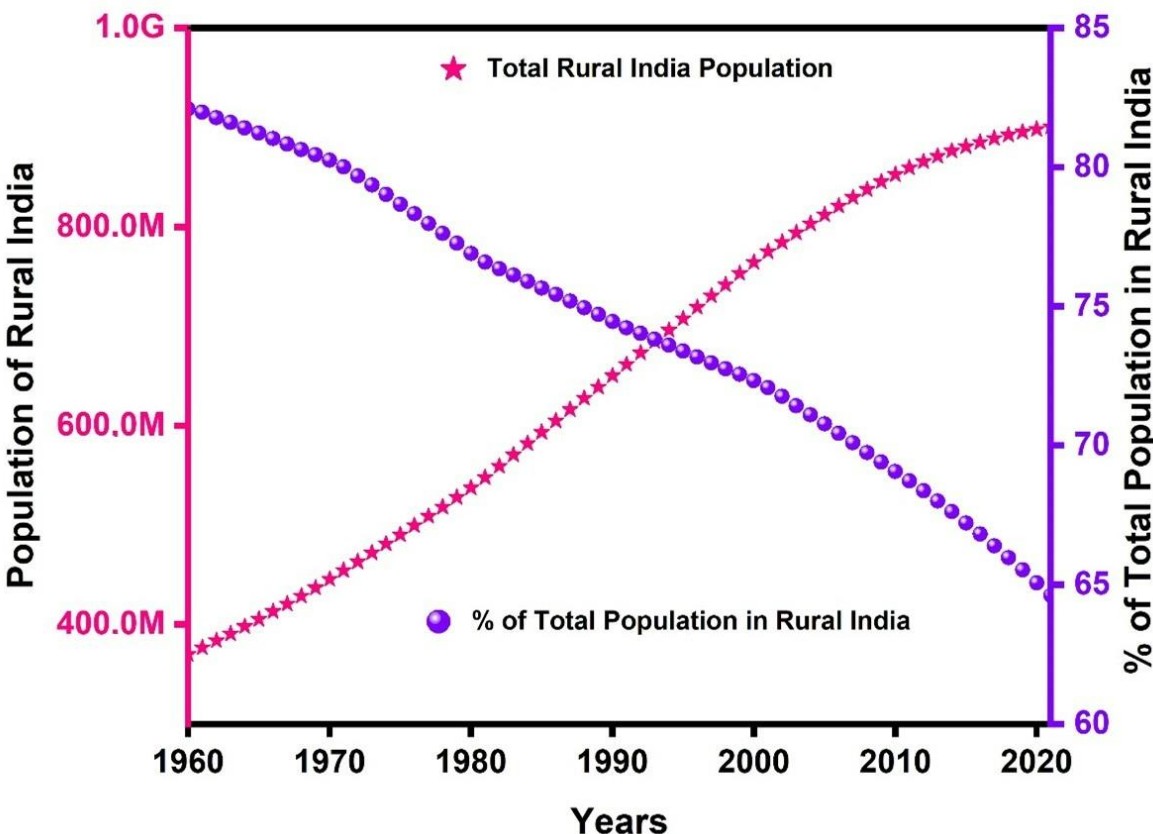

**Figure 1.** World Bank estimate: India's rural population growth as a function of years (left scale) ('M' = $10^6$ and 'G' = $10^9$), expressed as the percentage of the total population in rural India (right scale). The data used in this Figure 1 are extracted from the World Bank estimates based on the United Nations Population Division's World urbanization prospects: 2018 Revision [13,15].

Therefore, it is inevitable to reuse and recycle the GW and BW to meet the demand. Designing and constructing central sewage systems for treating the GW and BW, especially after the SBM, is a Herculean task, as it is virtually impossible to connect all the villages across India or even within its states. This article reviews the diverse methods that could be used for treating the BW generated from the lavatories constructed by the GOI through the SBM. These treatment processes provide micro-sized solutions pertaining to each locality and can solve the water scarcity issues of those villages. Currently, the BW generated in many rural areas of the world is discharged directly into the earth's surface, lakes, ponds, or underground through soak pits without treatment. This destroys the ecological balance of the environment and causes severe diseases to human beings. The methods discussed in this review would serve as a valuable guide for engineers and water management people in solving water scarcity faced by farmers to meet the growing food demands of the people, in particular in sparsely populated areas.

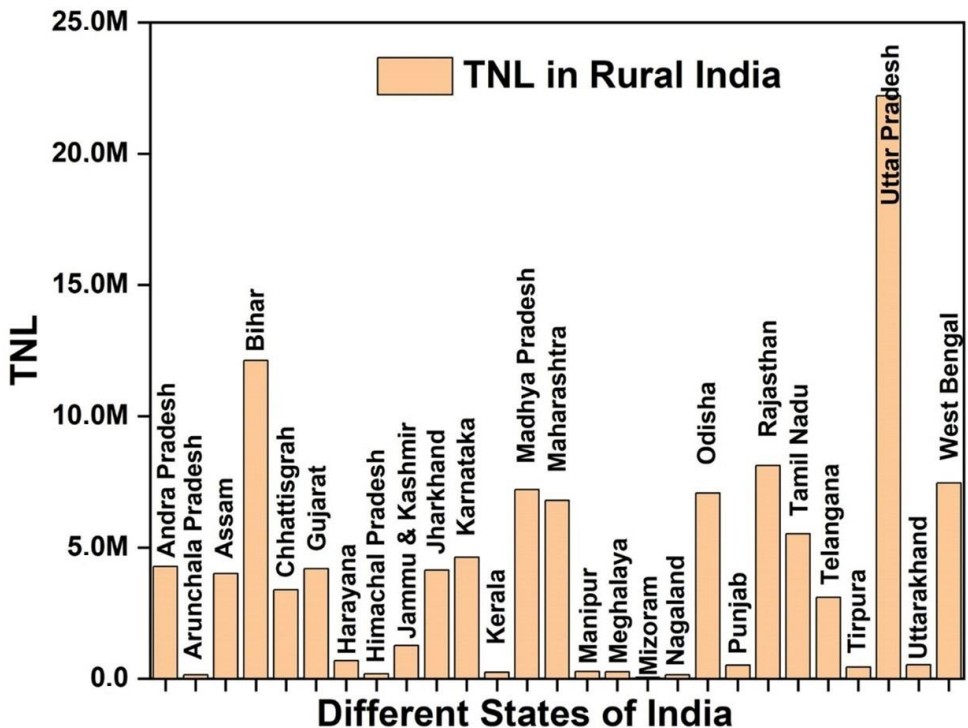

**Figure 2.** Bar chart depicting the total number of lavatories (TNL) built through SBM ('M' = $10^6$). Data for this chart are extracted from the Ministry of Jal Shakti of GOI [16].

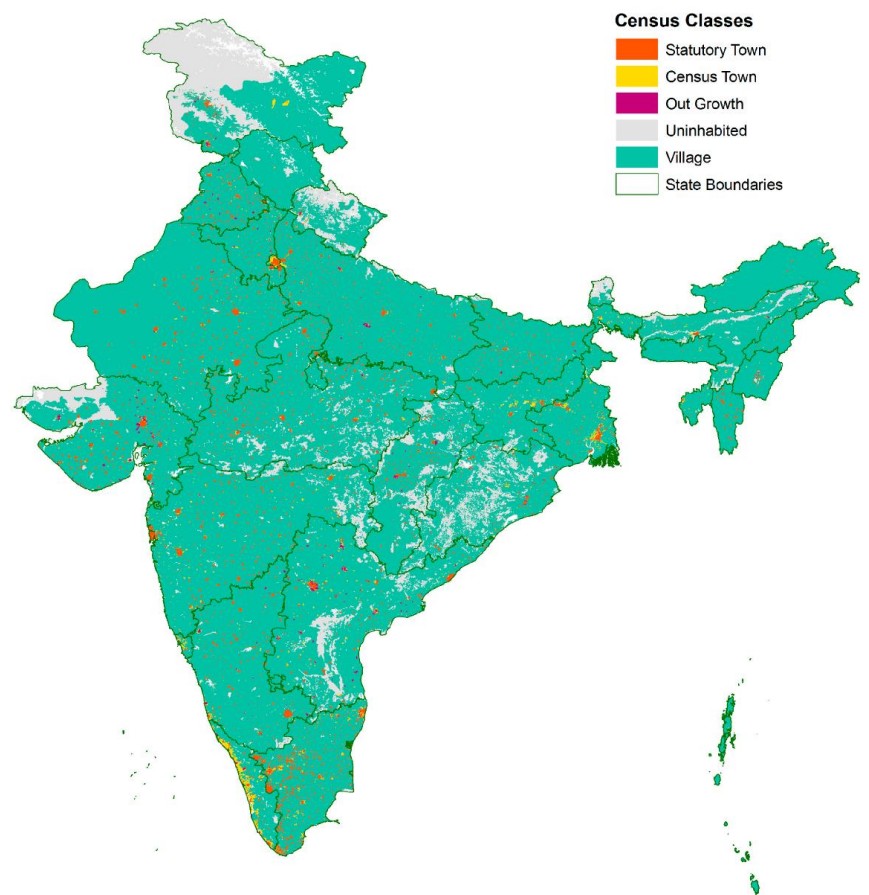

**Figure 3.** Rural and urban classifications based on the 2011 population census data of India, provided by India's Office of the Registrar General and Census Commissioner. This figure is adapted from the

work of Deborah Balk et al. [18]. Here, statutory town refers to the legally well-established urban area; census town refers legally to a rural settlement with a population greater than 5000 people [19]. An 'outgrowth' is a region of high population density positioned adjacent to a statutory town, which is on the verge of becoming a separate statutory town in the future [18].

## 2. Goal and Need of On-Site Treatment Processes

The establishment of a decentralized wastewater treatment process for the regions possessing low-density populations is a crucial move towards a community free from waterborne diseases. Such recycling-enabled societies attain water sustainability in the longer run even with the current climate change issues. The need for miniaturization of large-scale conventional wastewater treatment processes (like chlorination and ozonation) for such localized communities requires huge investment costs to ensure proper safety standards. Even if implemented, it may cause further exacerbation of problems, such as the generation of harmful byproducts after treatment, etc. These wastewater treatment plants require enormous energy consumption and are simultaneously responsible for a significant contribution to greenhouse gas emissions [20,21]. Other techniques like membrane filtration and UV irradiation are not suitable, as they involve high operational and maintenance costs [22]. Thus, easily operated, renewable, novel techniques are prerequisites for such decentralized areas.

A light-energy-based technique which possesses the potential to be driven by solar irradiation (renewable energy), the noble heterogeneous photocatalysis technique, has proven to effectively degrade different pollutants and microbes present in water [23–28]. However, so far, such an irradiation-initiated treatment technique has not been studied for the purification of BW from localized sources. Having said that, this article can be viewed as an abridgement of various novel, recently developed on-site BW treatment processes. The novel processes addressed here would serve as an efficient, economical, and ecofriendly solution for diverse remote rural areas in India where SBM has been implemented. The implementation of such ideas and the subsequent treatment of BW to produce potable or non-potable water would definitely empower the current Jal Jeevan Mission in India. While remote, sparsely populated rural India has been chosen as the exemplary target area throughout this paper, the local BW treatment options presented may also be more generally valid and applicable to other sparsely populated areas throughout the world.

## 3. Novel Diverse Techniques for the On-Site Treatment of Blackwater (BW)

BW generated from the toilet flush system contains a prominent quantity of pathogenic microbes (viruses, bacteria, and unicellular eukaryotes) produced by the products of human excretion. Thus, the untreated BW is always prone to cause diseases like diarrhea, cholera, dysentery, etc. [29]. At the same time, the untreated BW also possesses high quantities of nutrients (both micro- and macro-nutrients) as requisites for plant growth. These nutrients are primarily composed of nitrogen-, potassium-, and phosphorus-based materials. Interestingly, a component of human urine consisting of 75–90% nitrogen-based compounds is urea, and the rest comprises amino acids, uric acid, and creatinine, which can be directly taken up by plants [30].

Thus, the facile segregation of nutrients and microbial contents or the direct inactivation or removal of microbes, through proper treatment techniques, provides non-potable water for reuse. Usage of such nutrients from BW eliminates the need of hazardous synthetic chemical fertilizers and reduces the cost of current agricultural practices. The BW generated from localized sources, be it single-household toilets or community-based toilets, is mechanically separated into two phases, the overflowing liquid phase (supernatant) and the sediment, a solid phase. This review article will discuss the novel methods that can treat both phases in a decentralized manner, providing water and energy sustainability in the longer run. The entire picture of this work can be visualized in Figure 4. The methods

mentioned for BW treatment can also be implemented for treating GW or the mixture of both GW and BW.

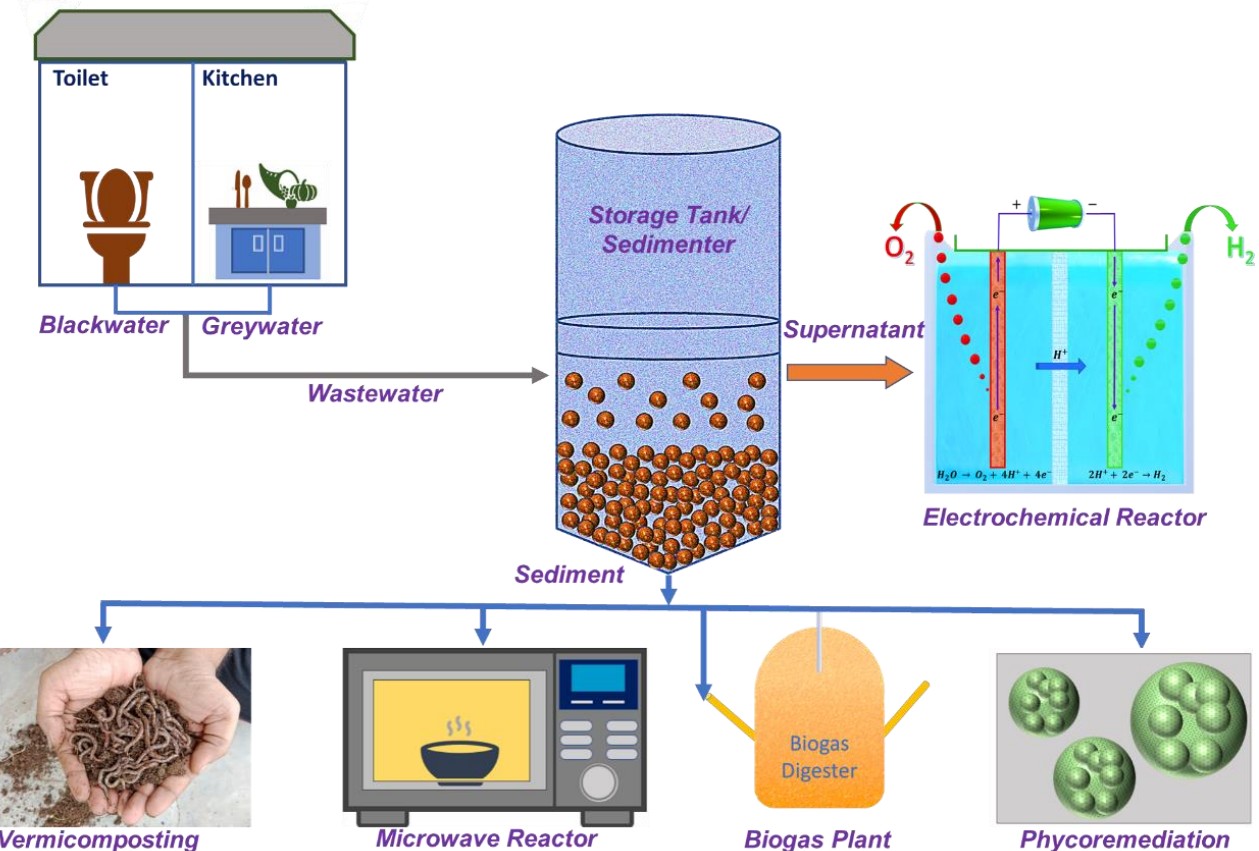

**Figure 4.** Diverse renewable methods proposed for the decentralized treatment of BW. These methods could also be applicable for the decentralized treatment of GW generated from rural areas.

*3.1. Electrochemical Treatment Process*

The electrochemical (EC)-based treatment process is the most multifaceted, easily integrable system for the on-site treatment of BW. It is potentially a feasible substitute for decentralized wastewater treatment for lavatories in single households or a group of households in a village. This methodology potentially eliminates the drawbacks associated with the scaling down of conventional treatment processes: (i) chlorination, (ii) ozonation, (iii) membrane filtration, and (iv) ultra-violet (UV) irradiation. The most crucial drawbacks are: (i) high operational and maintenance cost, (ii) generation of byproducts possessing acute side effects and further mineralization of such compounds, and (iii) safety issues associated with the storage of toxic chemicals required for the treatment [22]. The EC-based process does not require external chemicals, as it produces the chemicals needed for the treatment of BW in situ within the reactor, where the treatment is carried out. The basic principle of the EC method for the treatment process is the electrolysis of water, basically defined as the EC splitting of water into hydrogen and oxygen. The term EC indicates that the driving force for such splitting of water is the electric current applied to the system. The great advantage of this EC-based water treatment technique in its implementation in upcoming projects of the SBM is that the current required for treating the BW of a household in a village or an entire village's BW can be powered by solar energy collection systems. The space or the setup in which the above EC process occurs is called an EC reactor, electrolysis cell, or electrolyzer. This EC reactor consists of an anode and a cathode, and the space between these two electrodes is filled with a solution called electrolyte; in our case, the BW is the electrolyte. The reactor is separated by a semi-permeable membrane/diaphragm into two different compartments: the anodic and the cathodic compartments. The current

applied to the EC reactor is achieved externally using a power source (battery), as shown in Figure 5.

Conventionally in an electrolysis cell, which contains water as an electrolyte with no oxidizable organics (pollutants, microbes, etc.), the oxygen evolution reaction (OER) occurs in the anodic compartment, and oxygen gas ($O_2$) is produced. The hydrogen evolution reaction (HER) occurs on the cathodic side, and hydrogen gas ($H_2$) is produced, as shown in Figure 5.

In the presence of the environmental pollutants, in our case BW, instead of oxygen gas evolution, the direct heterogeneous (on the surface of the anode) or homogeneous elimination of pollutants and disinfection will take place in the BW electrolysis cell [31]. Such elimination occurs through mineralization (complete oxidation) of the pollutants. The advanced oxidation processes (AOPs) based on the above EC approach involve the generation of strong oxidizing agents originating from the reactions occurring on the anode side of the EC reactor, as shown in Figure 6. The electrolysis settings for BW treatment are controlled in such a way that the in situ direct oxidant generation on the anodic surface predominantly involves the generation of a hydroxyl radical (OH). This radical is a highly powerful oxidant with standard redox potential ($E^o$ ($OH/H_2O$)) of 2.8 V vs. the SHE (standard hydrogen electrode) [32]. OH has a very short lifetime in water in the range of $10^{-9}$ s (nanoseconds), and thus it is easily self-extirpated at the end of the treatment processes [33]. Subsequently, the reactive oxygen species (ROS), (i) hydrogen peroxide ($H_2O_2$), (ii) ozone ($O_3$), and (iii) superoxide anion radicals ($O_2^-$), are produced through the reaction between the hydroxyl radicals and molecular oxygen. These ROS-based oxidants play a major role in the purification (oxidation) of oxidizable pollutants and also disinfection of microbes present in the BW [34]. However, the primary disinfection is provided by reactive chlorine species (RCS), (i) active chlorine molecules ($Cl_2$, HOCl, $OCl^-$) and (ii) chlorine radical species (Cl). RCS generation is purely in situ within the EC reactor and has no need of external sourcing, as it is achieved through the reaction between natural chloride material in BW and hydroxyl radicals [35], as shown in Figure 6. The BW contains higher concentrations of chlorides, as the human urine itself contains a chloride concentration of about 50–150 mM [34].

Apart from the aforementioned oxidizing radicals, ROS and RCS, in EC-based AOPs, the pollutants are also directly oxidized on the anode surface by electron transfer; this is termed anodic oxidation or direct oxidation. In the case of an absence of pollutants, it is the simplistic water oxidation occurring on the anodic surface in the water electrolysis cell. From the perspective of degradation of pollutants present in wastewater from myriad sources, the anode chosen for EC-reactor-based purification should be a non-active electrode with very high oxygen ($O_2$) overpotentials. These non-active anodes possess the potential for $O_2$ generation in the range of 1.7–2.6 V vs. SHE, as these anodes produce weakly physisorbed (not binding covalently) hydroxyl radicals that have higher chemical reactivity towards pollutant oxidation, resulting in the mineralization of the harmful compounds [36]. Electrodes based on materials like tin dioxide ($SnO_2$), lead oxide ($PbO_2$), sub-stoichiometric titanium dioxide ($TiO_2$), and boron-doped diamond (BDD) are the notable non-active anodes. Amongst these options, BDDs are the most powerful anodes, producing a large quantity of hydroxyl radicals and exhibiting an excellent oxidation rate and enhanced current efficiency compared to other conventional electrodes employed for the degradation of pollutants in the regime of anodic oxidation [37]. However, the implementation of the above non-active anodes is not recommended for BW generated from single and/or community households, especially in decentralized treatment systems targeting the reuse of treated water or safe disposal of water into natural bodies. This is because the high overpotentials of anodes for $O_2$ generation during EC operation lead to the synthesis of highly stable perchlorate ($ClO_4^-$) and chlorinated organics (trihalomethanes (THMs) and haloacetic acids (HAAs)) in the treated water. This is due to the abundance of loosely bound hydroxyl radicals, and its mechanism of formation is shown in the inset box (red color) in Figure 6. These compounds are highly dangerous, inherently possess carcinogenic,

endocrine-disrupting properties, and deliver a potential threat to human beings and to their corresponding ecosystem [38,39]. Likewise, the non-active BDD and $PbO_2$ can also electrochemically generate other toxic oxidants like perphosphate ($P_2O_8^{4-}$), persulfate ($S_2O_8^{2-}$), and percarbonate $\left(C_2O_6^{2-}\right)$ through the anodic oxidation of its corresponding precursors in the electrolyte: phosphate, sulfate, and carbonate [37]. These strong oxidants along with perchlorate, even at very low levels, are very toxic and subsequently hinder the reuse of treated water through EC treatment.

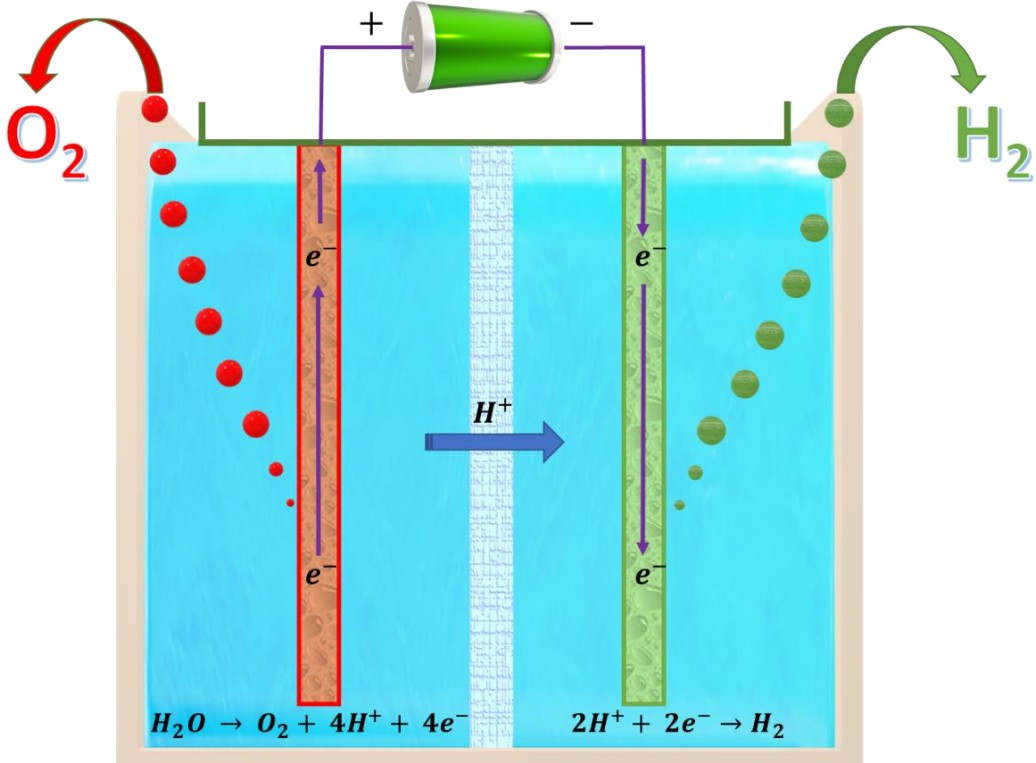

**Figure 5.** Water-splitting reaction occurring in a model electrolysis cell.

The facile formation of these strongly poisonous oxidants can be circumvented by the implementation of active anodes for the EC treatment of BW through anodic oxidation. Dimensionally stable anodes are materials composed of a titanium metal substrate coated with a thin layer of conducting iridium dioxide ($IrO_2$) or ruthenium dioxide ($RuO_2$); platinum (Pt), graphite, and bismuth (Bi)-doped titanium dioxide ($TiO_2$) [$BiO_x/TiO_2$] are some of the notable examples of active anodes. These electrodes are termed active anodes, as they exhibit low overpotentials for OER and are highly superior electrocatalysts for OER. Due to their low overpotentials for oxygen generation, the hydroxyl radicals generated on the above anode surfaces strongly interact with the anode's surface [37]. The production of such anode-surface-adsorbed hydroxyl radicals is shown in Equation (1):

$$MO_x + H_2O \rightarrow MO_x(OH) + H^+ + e^- \tag{1}$$

Due to the availability of higher oxidation states on the electrode surface of active anodes, the generated hydroxyl radicals are further oxidized into higher-state oxide. The resulting covalently bound oxygen species, termed "chemisorbed active oxygen ($MO_{x+1}$)", is oxygen present in the oxide lattice of the anode. This higher oxide formation reaction is shown in Equation (2):

$$MO_x(\cdot OH) \rightarrow MO_{x+1} + H^+ + e^- \tag{2}$$

This chemisorbed active oxygen ($MO_{x+1}$) along with the anode surface ($MO_x$) acts as a mediator in oxidation processes leading to the partial oxidation of organics. This prevents the formation of harmful disinfection byproducts (strong oxidants such as percarbonate, persulfate, and perphosphate) and to a certain extent chlorinated organics. Having discussed the nuances of the (electro)chemistry happening on the anodes' surface depending on the nature of the material chosen for the electrode in EC treatment, this review article will briefly address the current EC treatment systems studied for the safe recycling of BW. Recently, Xiao Huang et al. reported the practicability of a BW electrolysis cell (BEC) for the application of decentralized mobile toilet wastewater disinfection. The treated BW was aimed for reuse in toilet flushing and irrigation of crops [34]. Their study on EC disinfection efficiency was carried out for real BW collected during the continuous operation of a self-contained mobile toilet system, schematically shown in Figure 7.

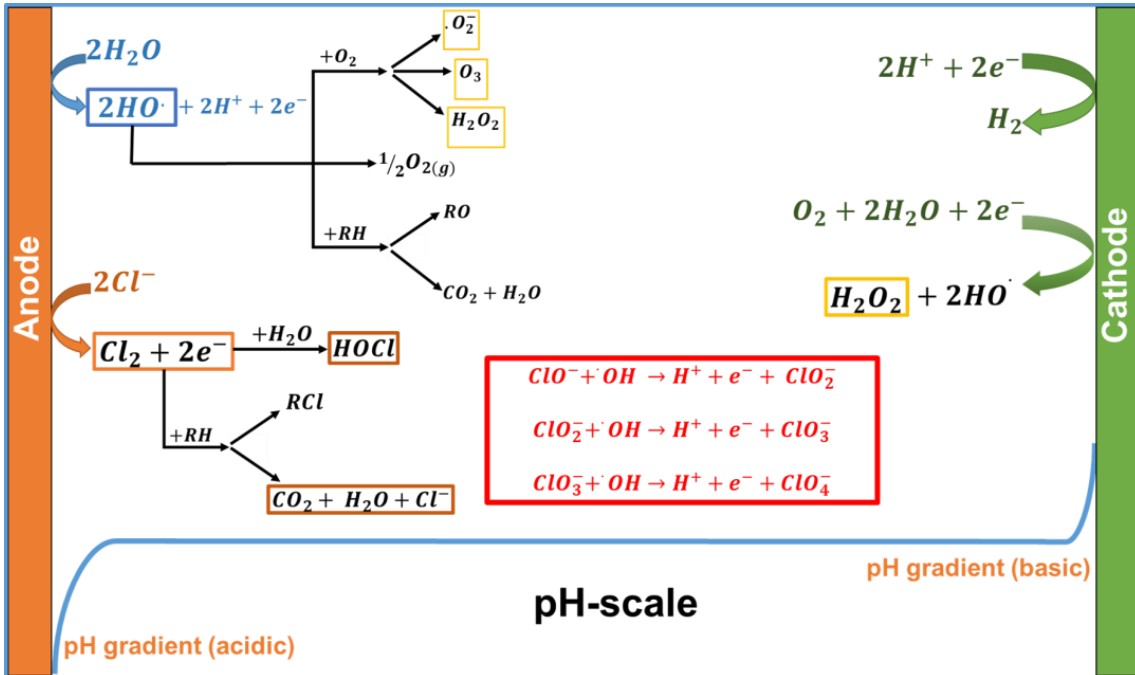

**Figure 6.** Oxidants generated in situ within the EC reactor during the EC treatment involving the elimination of organic compounds and disinfection of microorganisms present in BW. Hydrogen and other oxidants generated at the cathode side are indicated using the green-colored arrow. Apart from that, the rest of the oxidants are produced on the anode side.

This system was composed of an EC reactor, working inline in batch mode, treating 20 L of BW. The feed to this reactor is the supernatant from the sedimentation tank, which serves as an anaerobic digester. After treatment, the treated water returns to the clean water tank for reuse. However, BW from real toilet systems has a noticeable amount of microorganisms (both bacteria and viruses). The collected BW from the mobile toilet below (Figure 7) has a low concentration of microorganisms due to the truncated usage of the toilet, the presence of EC-treated water in toilet flush (which contains residual oxidants responsible for killing the microorganisms), and the long retention time of BW in the anaerobic digester (sedimentation tank). Thus, in their study, EC disinfection optimization was carried out on bench scale, and the collected real BW was seeded with fresh microorganisms. Their bench scale BEC possessed a working volume of 250 mL, composed of a $BiO_x/TiO_2$ anode with a very low overpotential for OER of about 0.32 V [40] and a stainless steel cathode. The distance of separation between the cathode and anodes was about 0.5 cm. The cell is operated in potentiostatic mode at 3 V, 4 V, and 5.5 V, resulting in current density values of 3.9, 1.2, and 2.4 mA/cm$^2$, respectively. Here, the inactivation of the microorgan-

isms was not observed under 3 V, but at an applied voltage of 4 V, microbial inactivation was achieved within 60 min of operation. The energy consumption for 1 h of EC reactor functioning was evaluated to be around 2 Wh/L, and for operating the entire setup in Figure 7, it requires around 13–15 Wh/L. This indicates that the BEC can be operated using commercial photovoltaic panels and represents a tremendous potential for a decentralized BW treatment process. However, at both 4 and 5.5 V, the free chlorine quantity (>1 mg/L) in the electrolyte was determined by the N,N-diethyl-p-phenylenediamine (DPD) colorimetric method. Due to the interference of chloramines and other oxidants ($[O_3]$, $[H_2O_2]$, $[ClO_2]$, etc.) in the DPD method, the observed quantification of free chlorine could be an overestimation. It is important to stress that quantification of individual oxidants is also virtually impossible in real BW due to its complexity of composition and swift reactions of oxidants with organic matter present in the BW. Hence, in this study, electrolysis studies were also carried out on model BW containing no chlorine $[Cl^-]$ and at an applied voltage of 4 V; less than 0.5 mg/L of total oxidants was observed. This EC disinfection was compared with the traditional chemical chlorination (CC) technique; here, the EC disinfection outperformed the disinfection by the CC technique, even at the highest dosage level of $Cl_2$ (36 mg/L). For both treatments, the observed composition of halogenated organics generated was similar. A negligible quantity (<5%) of brominated organics was observed in the treated water samples. Nevertheless, the quantity of disinfection byproducts (like chlorinated organics) was significantly higher in EC-treated BW than in the CC-treated BW. The concentration of the chlorinated organics was well within the permissible range for EC-treated BW and was similar to the range observed in chlorine-disinfected wastewater [41,42].

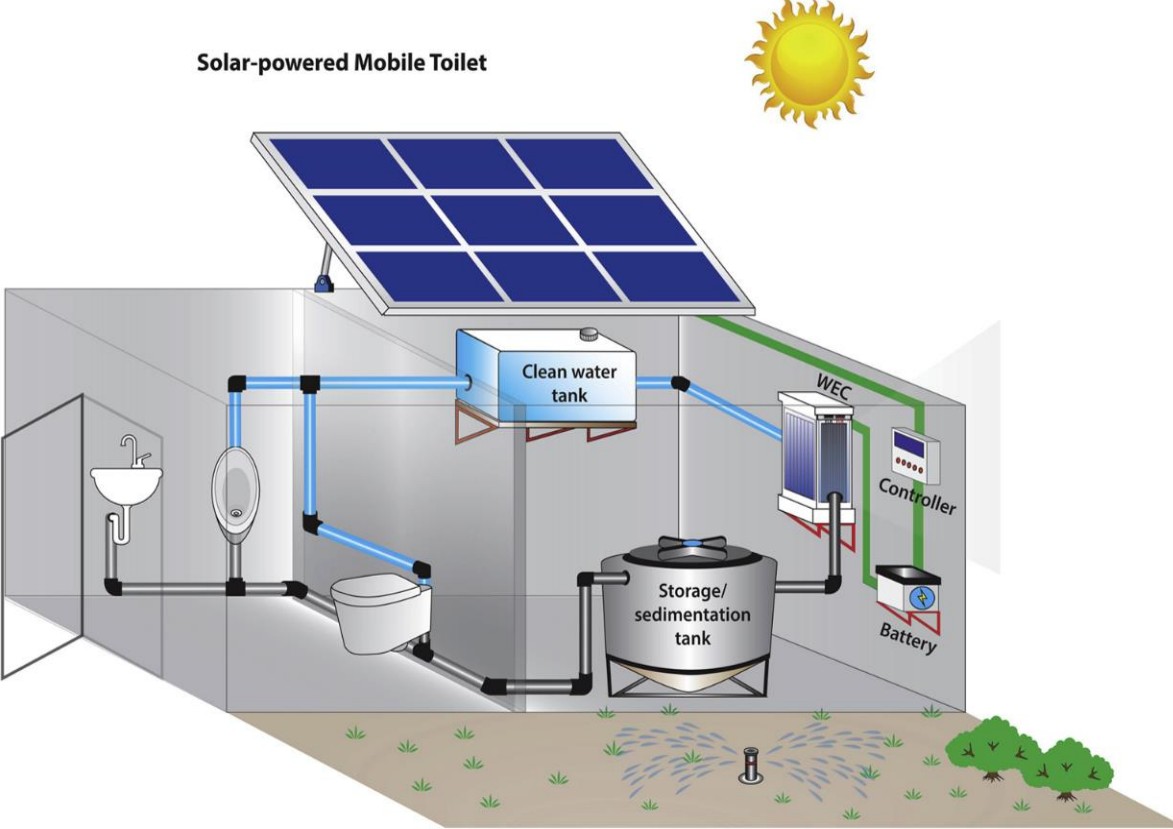

**Figure 7.** Schematic representation of the mobile toilet system employing the electrolysis cell powered by the solar panel for the treatment of BW, reported by Xiao Huang et al. Reused under the Creative Commons license [34].

The estimated total chlorine concentration of EC-treated (4 V for 60 min) BW was equivalent to CC-disinfected Blackwater with a dosage of 5 mg/L of $Cl_2$. Finally, increasing

the operating voltage from 4 V to 5.5 V doubled the generation of chlorinated organics, indicating that apart from the nature of the anode materials, the operating voltage has a profound influence on the disinfection byproducts generated in the treated water. The conclusion from this study clearly shows us that applying higher voltage to the electrolysis cell and extended reaction time should be avoided when the disinfection of the BW is completed. During the disinfection of BW, microorganisms (especially bacteria) are inactivated either by the complete destruction of the genome integrity of their cells or by the damages to the cell wall. Thus, the maintenance of chlorine residues (2 mg/L) in the treated water tank (clean water storage tank in Figure 7) is recommended. As such, the presence of chlorine will limit the regrowth of bacteria in the EC-treated BW, and the health risks associated with non-potable water reuse are completely eliminated.

Guruprasad et. al. recently reported a synchronized (wetland + EC reactor) system for the localized treatment of BW from single-household toilet (SHH) with quartet usage [43]. The EC reactor was composed of an active iridium-oxide (IrO$_x$)-based mixed-metal oxide (MMO) anode, stainless steel cathode, and cationic exchange membrane (CEM). The distance between the anode and cathode was around 5 mm. The CEM was supported by the frame within the EC reactor. The frame structure extends below the EC reactor into the buffer tank attached with baffles, preventing dead zones for disinfection, schematically depicted in Figure 8. The CEM divides the EC reactor into two equivalent chambers (anode and cathode side) with an 8.5 L volume each, and the buffer tank has a volume of 100 L.

The BW from the continuous operation of SHH flows into the septic tank, then into the vertical subsurface of a constructed wetland with a surface area of 0.5 m$^2$/PE (population equivalent) planted with Canna indica plant. The effluent from the wetland is sent to the buffer tank, into the EC reactor, then finally discharged. The wetland was operated continuously for 5 days prior to integration with the EC reactor, with an influent flow rate of 200 L/day. The wetland retains the organic matter in the BW and degrades it aerobically. However, such a wetland was unable to eliminate the bacterial pathogens completely. After the integration with the EC rector, the log reduction of bacterial coliform was 6.0 ± 0.4. Electrolysis of the BW was carried out in constant current mode (4.4 A) with an energy consumption of 16.7 Wh/L. Inside the EC reactor, the BW first flows through the anodic chamber and then through the cathodic chamber. The continuous flow rate of BW from the septic tank to the integrated system was 180 L/day. The wetland reduces the organic and mineral load in the BW, enabling the continuous function of the EC reactor for 60 days, after which the EC reactor was dismantled, and the membrane was washed to remove the accumulated organic layer on its surface [43]. In this trial period, the average depletion of total ammoniacal nitrogen (TAN), chemical oxygen demand (COD), total Kjeldahl nitrogen (TKN), and orthophosphate concentration due to the combined effect (wetland + electrolysis) was 84 ± 8 %, 84 ± 7%, 45 ± 14%, and 98 ± 1%. Within the EC reactor, the COD pruning is due to the direct or indirect oxidation of organic matter in the anode chamber; TAN reduction is due to myriad reasons, one of which is the migration of NH$_4^+$ ions to the cathode chamber and the subsequent bubbling of ammonia [43].

In view of augmented global population growth accompanied by a growing energy and food demand, the recovery of scare nutrients (for example, phosphorus) from the BW is highly advantageous as compared to its degradation by oxidation [44]. Within the EC field, such selective recoveries of nutrients are achieved through electrodialysis (ED) employing the usage of ion-selective membranes. However, such recoveries are not suitable for BW generated from localized sources (such as lavatories built under SBM) and/or wetland due to the low concentration of nutrients [45].

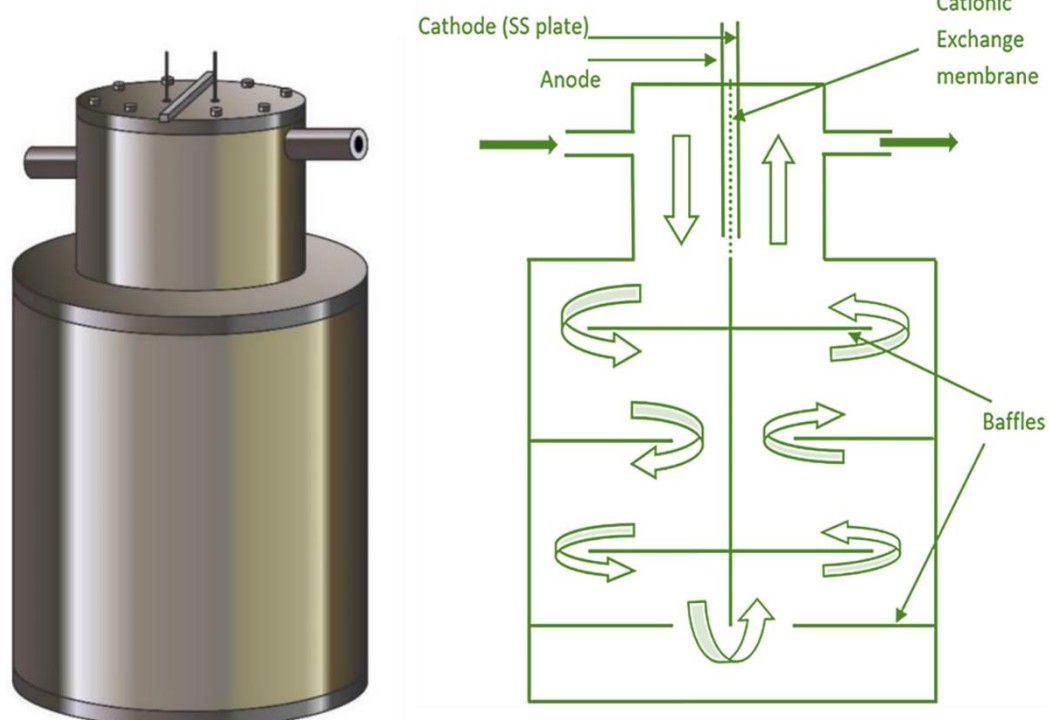

**Figure 8.** Schematic and cross-sectional diagrammatic representation of single-household EC setup showing the upper electrolysis cell and the bottom baffle-fitted hold tank, as reported by Guruprasad et al. Reused under the Creative Commons license [43].

To conclude, the sludge-free liquid phase of BW can be easily treated (disinfected) with an EC reactor powered by sustainable power supply technology (solar panels). These systems are viable for the decentralized treatment of BW. The successful direct continuous treatment of BW using small EC reactors inspires and encourages us to propose water electrolyzers as an upgraded system for the current "Retrofit To Twin Pit Abhiyan" launched in October 2022, carried out under the SBM by the GOI. Here the pits serve as anaerobic digesters, and liquid flowing out of the pits can be disinfected with EC reactors.

Having said that, the composition of the BW and the nature of the pathogens present in such BW vary from different localities. These factors depend heavily on the food habits, lifestyle, and climatic conditions of each locality. Hence, preliminary studies targeting the composition of BW in various locations far away from the centralized treatment systems, as well as optimization studies of EC reactors with highly efficient electrode materials reducing the energy consumption while at the same time maintaining the disinfection efficiency in the upcoming years, would provide pathways for the easy reuse of treated BW in India and also in other parts of the world focusing on the decentralized treatment of BW. In the upcoming sections of this review article, technologies developed for the handling and treatment of solid waste generated from BW are discussed.

### 3.2. Vermicomposting

Vermicomposting is a process in which earthworms convert the solid organic waste into manure rich in highly nutritional content. Vermi-processing toilets have a design in which human excreta are treated by earthworms and redworms. In this process, human waste is transformed into cake-like structures called vermi-cakes by earthworms. Vermi-cakes are used as manure and fertilizers [46]. Recently there has been a lot of emphasis on organic fertilizers in lieu of chemical fertilizers. In this method, the organic matter, such as cow dung, kitchen waste, and farm residues, is consumed by earthworms and ejected in a digested form called worm casts and popularly known as black gold. The casts are rich in nutrients, growth-promoting substances, and beneficial soil microflora, and they

inhibit pathogenic microbes. Vermicompost is a stable fine granular organic manure which enriches soil quality by improving its physicochemical and biological properties. It is highly useful for raising seedlings and for crop production. Recently, it has become popular as a major component of organic farming. Recent research has indicated that vermicomposting can be useful in other fields, such as wastewater treatment, reduction in BOD and COD, soil remediation, and energy production [47–49]. By utilizing vermicompost to produce energy from waste and promoting the three R's (reduce, recycle, and reuse products and resources), vermicomposting plays an invaluable role in a country's circular economy. To ensure global sustainability and circular economies without harming the environment, vermicomposting technology must be considered as a viable management technique. The GOI has also promoted the adoption of biological techniques such as composting and vermicomposting for waste recycling [50].

There has been a lot of emphasis on treating organic waste and stabilizing sludge from wastewater treatment plants using vermicomposting and vermifiltration [51]. This article provides information on the design, processes, and uses of vermifiltration technology. Vermicomposting toilets are becoming increasingly popular, in which earthworms break down human feces, urine, and toilet paper. The system includes a conventional flush toilet and handles waste on site. This methodology uses less water and converts dry human feces into humus (organic matter) in a pollution-free manner. The vermicompost toilet comprises two main parts, namely the conventional flush toilet and the worm tank. The worm tank is where BW is treated by worms, schematically depicted in Figure 9.

An intermediate bulk storage tank is used to store the sewage coming from the flush toilet before it is sent for treatment in the worm tank. The tank is then encased in insulated housing (see Figure 9) to prevent the worm tank getting too hot in the summer (cold winters may slow down activity but are less likely to kill the worms than a hot summer). It is critical that the worm tank drains the toilet flush water to maintain a healthy compost worm ecosystem. In order to permit the free flow of the liquid, it is necessary to arrange internal pipes with perforations and a gravel bed at the bottom. Typically, the tank is filled with $\frac{3}{4}$ coarse organic material and a small layer of kitchen scraps, with partially finished compost or manure at the top. It would take approximately 90 days to convert all the feces into usable decomposed organic matter suitable for agriculture. High-yield worms can be sourced from agricultural universities or online suppliers, and the number of worms required for the biodegradation depends on the amount of the biodegradable material. The wash water (supernatant) can be treated using an EC method discussed in this article (see Section 3.1). The average values of the composition of raw wastewater (influent) and treated water (effluent) are shown in Table 1.

The studies indicated that the effluent quality of vermicompost-treated samples showed significant removal of BOD (86.3%), COD (70.2%), and suspended solids (SSs) (45.7%). The significant removal of BOD is due to the enzymatic action of earthworms that degrade the high concentration of organic matter. A removal of 80% suspended solids was observed, which is attributed to the fact that earthworms digest the solids and also improve the adsorption properties of the sands and soils by their grinding action. Similarly, the removal of Total Coliforms (TC) (55.74%), Fecal Coliforms (FC) (53.97%), and Fecal Streptococci (FS) (60.36%) to levels considered acceptable for recreation and irrigation indicates that vermifiltration products are pathogen-free [52]. This suggests that the substantial reduction in pathogens in vermifiltration is due to antimicrobial activity in the celeomic fluid of earthworms and in the microflora associated with vermifiltration [53].

In essence, the current reports about vermicomposting technology reveal that pathogen removal is a significant benefit in addition to organic fertilizer. Vermifiltration is a cost-effective technology for sewage treatment that is highly efficient, convenient, and has the potential for decentralized treatment. It can be easily integrated with the lavatories constructed under the SBM. Vermifiltration technology is an excellent microbial–geological system and can be improved to offer a greener planet in the future.

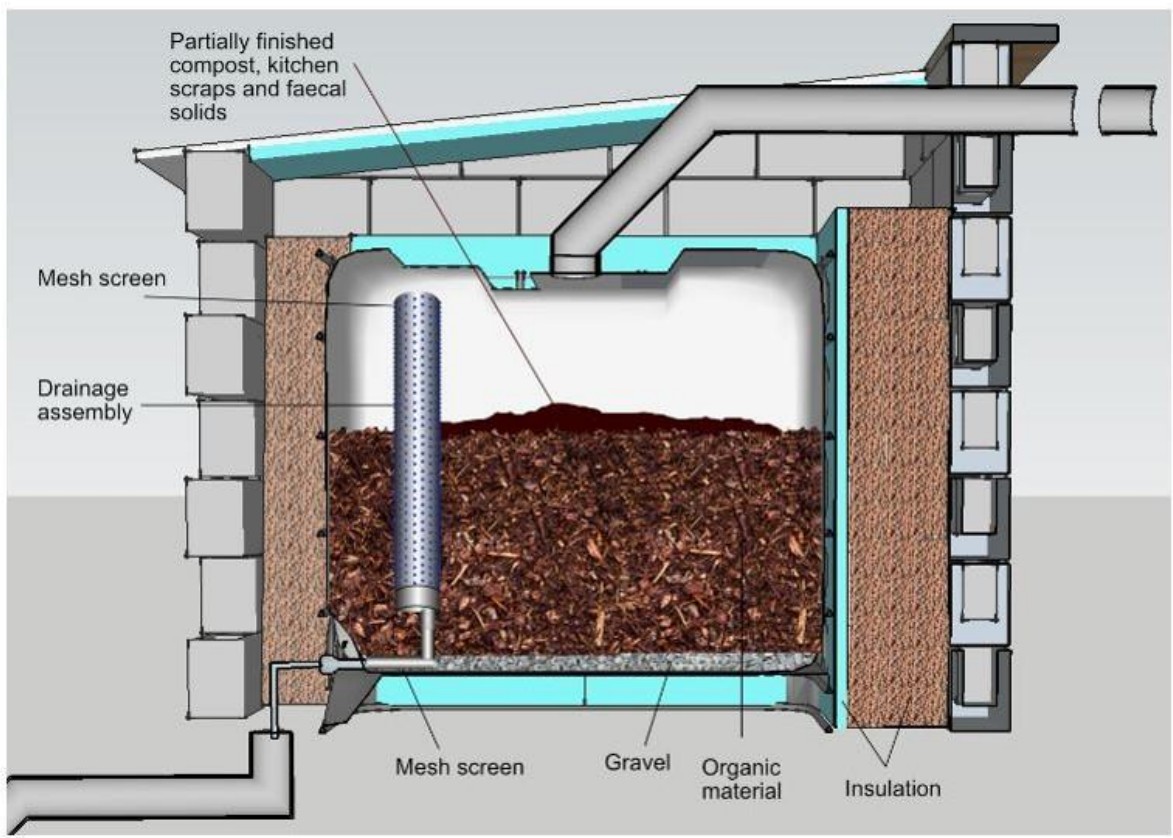

**Figure 9.** Cross-sectional depiction of worm tank in vermicompost-based treatment system integrated with single-household toilets. Reused under the Creative Commons license (CC BY-NC-SA 4.0) [54].

**Table 1.** Comparison of influent and effluent characteristics of the sludge. Data mentioned here are procured from the work of Sudipti Arora et al. [55].

| Parameters | Influent | Effluent | Percentage Reduction (%) |
|:---:|:---:|:---:|:---:|
| BOD (mg/L) | 260 | 35.5 | 86.3 |
| COD (mg/L) | 345.8 | 102.8 | 70.2 |
| SS (mg/L) | 252.3 | 137 | 45.7 |
| TC (MPN/100 mL) | 7.05 | 3.12 | 55.74 |
| FC (MPN/100 mL) | 5.54 | 2.55 | 53.97 |
| FS (MPN/100 mL) | 6.03 | 2.39 | 60.36 |

*3.3. Microwave (MW)-Based Reactor*

Recently, the MW-based thermal treatment method has gained importance in treating BW sludge, owing to its rapid heating [56,57]. Here, the sludge samples are exposed to microwaves produced by a microwave electron tube known as a magnetron. These microwaves cause the water molecules in the sample to vibrate, producing thermal energy and thereby heating the sample in the least possible time [56,58–60]. Here, MWs are able to reduce the sludge volume by about 70% while keeping the pathogen concentrations below analytical detection limits under experimental conditions [57,61]. In a study reported by TU Delft, a domestic MW oven (Samsung, MX245, Samsung Electronics Benelux B.V., The Netherlands) was employed for the treatment of BW collected from the decentralized

sanitation site at Sneek city in the Netherlands [57]. The effect of MW treatment on the sludge volume reduction and pathogen removal is schematically shown in Figure 10.

The MW methodology with the addition of 3 wt.% hydrogen peroxide ($H_2O_2$) to the sludge ensures all the COD present in the sample is converted to a soluble form, which in turn enhances biodegradability and methane production. This method also offers the advantage of non-contact heating with a high degree of uniformity, rapid reduction in volume, and a compact and portable nature with low thermal energy loss [60,62]. The efficiency of the MW increases with the power and the contact time of MW radiation. The extra power contributes to the rapid vibrational and rotational movement of water molecules. This movement in turn releases extra heat, resulting in the evaporation of water molecules and the decomposition of the various contaminants in the sludge [57,60]. Specifically, the MW radiation brings about the intensification of the degradable material in the BW sludge compared to conventional heating, such as with an electric furnace. Owing to its unique capabilities of rapid heating, MW technology appears very promising for situations requiring the rapid treatment of fecal sludge (FS) matter with reduced carbon and reactor footprints [56,57].

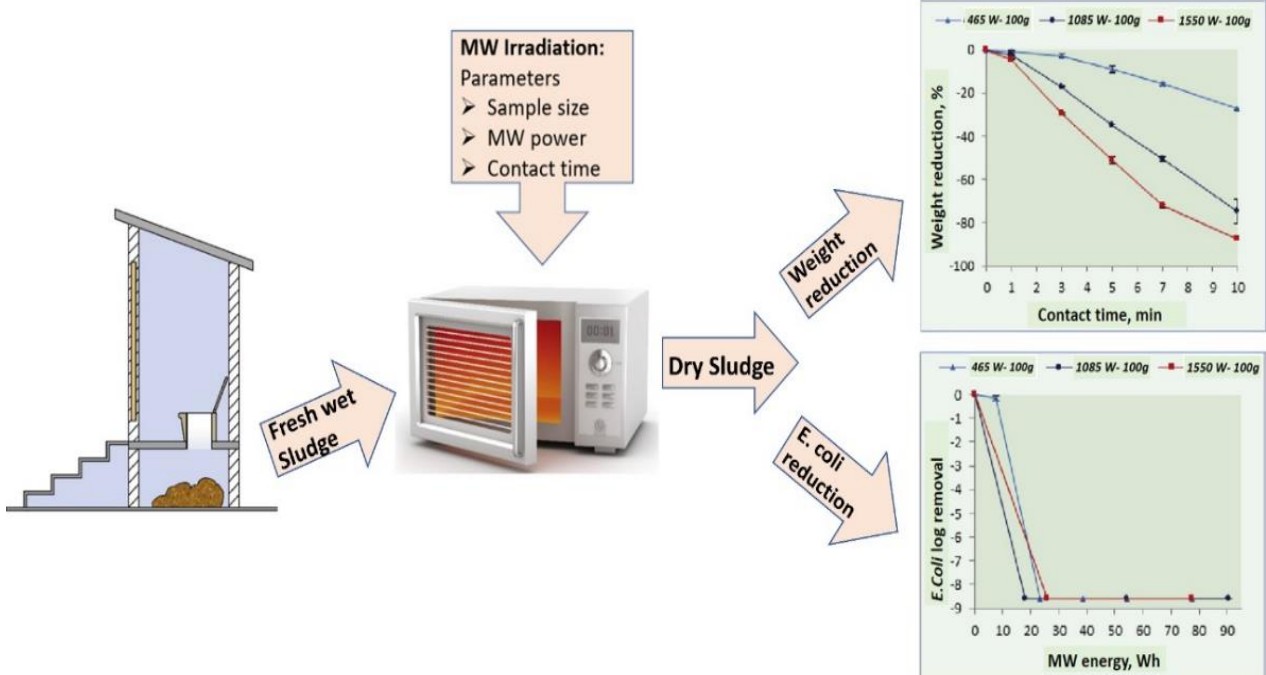

**Figure 10.** Diagrammatic representation of the process implementing the usage of domestic MW for the reduction in BW sludge volume and the pathogen reduction in the treated sludge. The above figure was adapted from the work of Peter M. Mawioo et al. Reused under the Creative Commons license agreement (CC BY 4.0) [57].

Researchers recently succeeded in using an MW reactor at pilot scale to treat 4 kg of sludge with MW radiation for time durations ranging from 30 to 240 min [63]. The schematic diagram of such a scaled-up MW reactor is shown in Figure 11. Here, the sludge with a uniform thickness of 0.5 cm was exposed to MW irradiation at 3.4 kW, and the temperature was raised to 102 °C. The treated samples were analyzed for the reduction in volume, calorific value, organic matter, nitrogen, phosphorous, and Escherichia coli. This study showed complete bacterial inactivation and a sludge weight/volume reduction above 60%. Moreover, the dried sludge and condensate had high energy ($\geq$16 MJ/kg) and nutrient contents (solids: TN $\geq$ 28 mg/g TS and TP $\geq$ 15 mg/g TS; condensate: TN $\geq$ 49 mg/L TS and TP $\geq$ 0.2 mg/L), having the potential to be used as biofuel, soil conditioner, fertilizer, etc. [63].

Thus, the MW reactor can be applied for the rapid treatment of sludges generated from areas lacking the capabilities for agriculture and rainfall, like desert areas in the world. Chemically bound heavy metal ions are the most toxic and difficult to remove from waste sludge. MW technology decomposes these metal ions quickly, and the treated sludge can then be used as organic fertilizer [60,64]. To conclude, MW treatment of BW and municipal wastewater is of significant interest primarily because of the rapid heating. This methodology lacks proven global technologies and requires verification and economical analysis for practical implementation.

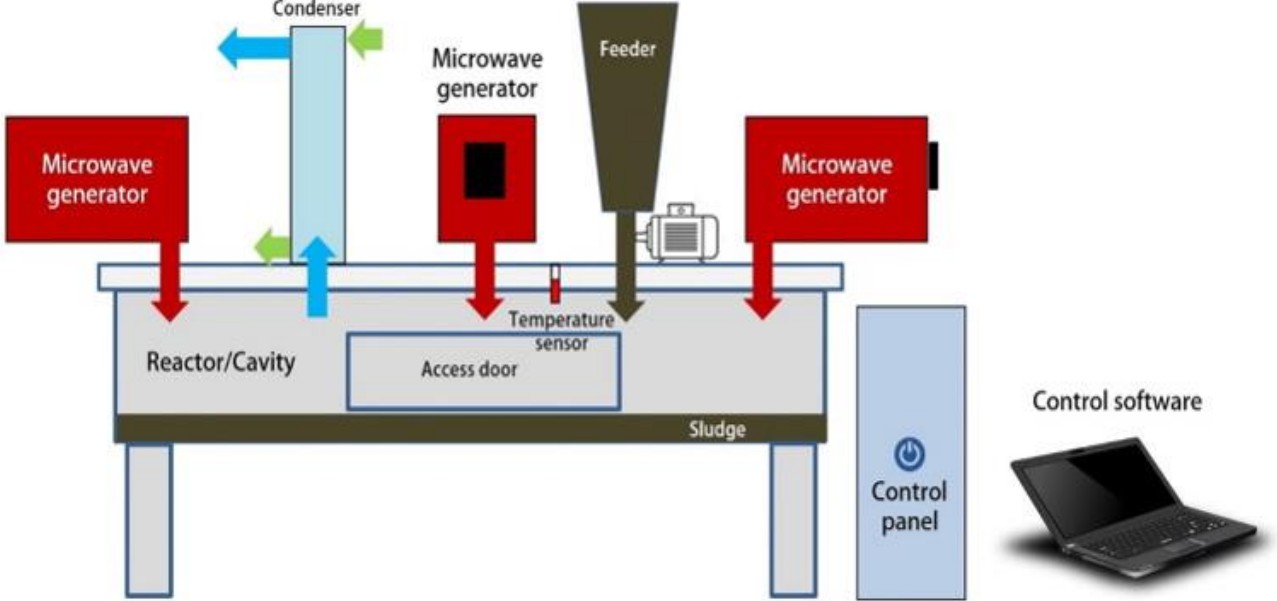

**Figure 11.** Schematic representation of pilot plant-scale MW reactor for the treatment of sludge from BW. This figure was reused from the work of Peter M. Mawioo et al. Reprintedunder the Creative Commons license (CC by 4.0) [63].

*3.4. Biogas-Based Treatment*

The term biogas means a mixture of methane and carbon dioxide produced by the bacterial decomposition of organic wastes, and it is used as a fuel. The disintegration of organic waste by anaerobic digestion has gained global attention due to its significant environmental and economic benefits. This technique reduces local waste by recycling. This in turn conserves resources, reduces greenhouse gas emissions, and builds economic resilience in an uncertain future for energy production and waste disposal [65]. Biogas would replace fossil fuels to meet the current energy needs of industry and the transportation sector, which in turn contributes to a greener societal ecosystem. $CO_2$ production is significantly less for biogas when compared to conventional fuels. It can be used as fuel in cooking, a source for heating and lighting, and can be used to generate electricity [66]. There are a number of other advantages to biogas plants. They are easy to install and do not require any sophisticated equipment. Furthermore, the costs of installation and handling are cheap. These plants can be installed in villages for individual houses or for a group of houses. The digestate that comes out from the plant can be used as fertilizer in agriculture as a substitute for harmful chemicals.

Water, protein, undigested fats, polysaccharides, bacterial biomass, ash, and undigested dietary remains are the main components of feces in the BW. As a proportion of wet weight, the primary elements in feces are oxygen (74%), hydrogen (10%), carbon (5%), and nitrogen (0.7%), including the oxygen and hydrogen found in the water portion of the feces. Feces are made up of approximately 75% water and 25% solids. Organic materials make up 84–93% of the solid portion. A calorific value of 4115 kcal/kg of dry solids can be used as the design standard for the calorific values of feces [67].

Human feces can be a viable source for biogas production. The slurry from BW after coagulation can be used as a substrate for biogas production or can be used along with other household organic waste such as kitchen waste and agricultural waste for the production of biogas. Table 2 presents a comparison for various organic sources in terms of biogas produced per kg of dry waste [68].

**Table 2.** Comparison of diverse raw materials and the corresponding biogas yielded from those sources [68].

| Source | Waste Amount/Day/kg | Water (%) | Dry Matter (%) | Biogas $m^3$/kg Dry Waste |
|---|---|---|---|---|
| Cow | 20–30 | 80 | 20 | 0.023–0.040 |
| Rooster/Hen | 0.15–0.20 | 72 | 28 | 0.065–0.116 |
| Human | 0.10–0.40 | 77 | 23 | 0.02–0.028 |

The production of biogas works on the principle of anaerobic digestion of organic matter in the absence of oxygen by the action of certain anaerobic bacteria, and this process takes place in four steps, namely: (i) hydrolysis, (ii) acidogenesis, (iii) acetogenesis, and (iv) methanogenesis [69].

*Hydrolysis*: Here, proteins, carbohydrates, and lipids (fats) are digested by hydrolyzing bacteria into liquid monomers and polymers in the first stage, after which they are converted back into amino acids, monosaccharides, and fatty acids, respectively. *Acidogenesis:* The soluble organic monomers of sugars and amino acids are converted by acidogenic bacteria to ammonia, ethanol, acids (such as propionic and butyric acid), acetate, $H_2$, and $CO_2$ in the second stage. *Acetogenesis:* In this step, acetogenic bacteria convert long-chain fatty acids, volatile fatty acids, and alcohols into hydrogen, carbon dioxide, and acetic acid. *Methanogenesis:* At this stage, methanogenic bacteria transform the hydrogen and acetic acid into methane gas and carbon dioxide. Here, gaseous impurities such as hydrogen sulphide, nitrogen, oxygen, and hydrogen also appear along with methane and carbon dioxide. The higher percentage of methane (>45%) makes the biogas suitable for combustion, i.e., has higher calorific value of the fuel [69].

Biogas plants are available in sizes ranging from 0.5 $m^3$ to 20,000 $m^3$, which are available for domestic, community, institutional, industrial, and commercial applications. Conventionally, the different types of biogas plants are: (i) fixed-dome plants, (ii) floating-drum plants, (iii) balloon plants, and (iv) tubular-design-based digester plants. The classification of biogas plants is mostly based on the type of the digester employed for their operation. The floating-drum plants are the ones that are highly recommended because of their universal applicability and easy maintenance, and such a plant is shown in Figure 12. In the past several decades, several small-scale floating-drum biogas plants were constructed in various rural areas in India, and they were popularly called Gobar gas plants. A typical biogas plant consists of a reception tank, digester/fermenter, gasholder, and overflow tank. The reception tank collects organic matter that needs to be digested; the digester is the part where the anaerobic digestion of the feces takes place. Methane along with carbon dioxide and other gases are produced in the digester. In the floating gas holder, the obtained gases are collected and then let outside for use. An overflow tank serves the purpose of discharging the processed sludge. The gas produced in the digester in the first few weeks is mainly carbon dioxide ($CO_2$) [70]. The gas is not flammable and can be released into the atmosphere. Following this, the flammable methane content of the gas starts increasing ($CH_4$ > 45 Vol %) and can be used as fuel. The leftover sludge can be used as a source of nutrients for agriculture. The production of biogas using human waste can provide various benefits in terms of the environment, as a soil nutrient, and as an alternative source of energy. To conclude, we highly aspire towards and envision that the mutual concoction of EC reactors and floating-drum biogas plants (anaerobic digesters) would provide the pathway for the decentralized treatment of BW in locations with highly dispersed populations.

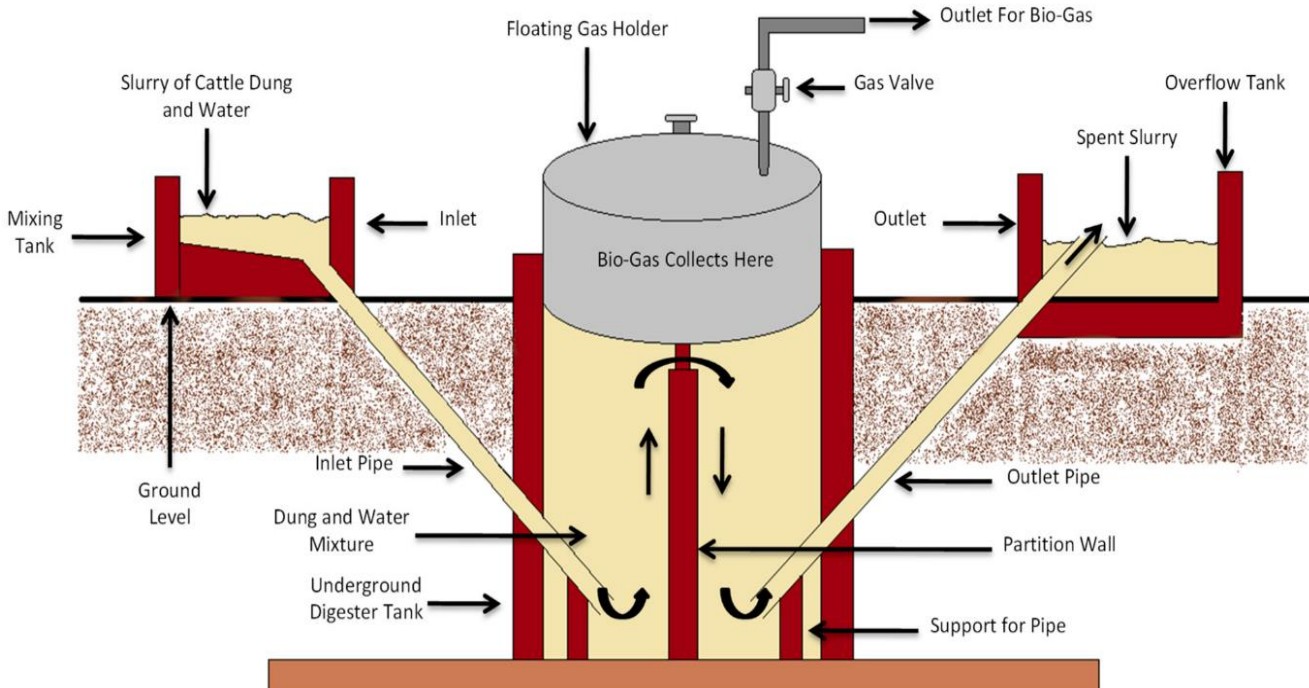

**Figure 12.** Schematic representation of the typical floating-drum biogas plant. This diagram is adapted from the work of Prachi Pandey et al. Reused under Creative Commons license [71].

*3.5. Phycoremediation*

The term phycoremediation means the usage of algae to remove, degrade, or biotransform undesirable substances in wastewater [72,73]. Alternatively, phycoremediation can be defined as the employment of micro- and macroalgae for the removal or biotransformation of toxic pollutants, including nutrients and xenobiotics, from wastewater [74]. Recently there has been a lot of attention by researchers on treating wastewater using biological systems, i.e., micro- and macroorganisms, in lieu of traditional expensive chemical treatment methods. This process involves the acceptance of contaminants from polluted bodies of water by algal cells followed by degradation into nontoxic forms and sometimes nutritional sources [75]. Here, pollutants are decontaminated by a method known as biosorption, where pollutants simply bind to the cell walls of the algae. In another technique called active uptake, algae have the potential to absorb and use these toxic pollutants in their cellular metabolic processes, thereby rendering the pollutants non-toxic.

The remediation of the pollutants with the aid of algae has proven ideal because it is eco-friendly, cost-effective, and easily managed. Microalgae form peptide bonds during photosynthesis that can bind toxic metal ions from wastewater and form organometallic complexes. These complexes are then further sequestered into the vacuole, in turn neutralizing the toxic effect of the metal ions. For instance, the macroalgae Caulerpa lentillifera have the potential to remove the metals copper, cadmium, lead, and zinc through biosorption by some specific functional groups present on the cell surface [75–77].

Algae exist everywhere in aquatic ecosystems and are adapted to diverse conditions. This has enabled algae to develop a wide range of resistance to natural surroundings [78]. Due to this benefit, algae have been widely used in the bioremediation of pollutants, producing clean water as well as useful biomass that may be used as feedstock for a variety of valuable goods, including food, feed, fertilizer, pharmaceuticals, and more recently, biofuel. When paired with the production of biofuel and integrated with waste treatment, algae have the potential to be a carbon sink by removing carbon dioxide through photosynthesis [79]. A wide range of microalgae, such as Chlorella, Scenedesmus, Phormidium, Botryococcus, Chlamydomonas, and Spirulina, for treating domestic wastewater has been

found to be effective and encouraging [80]. Specifically, Chlorella vulgaris was found to be effective in treating BW [81]. This system has the potential to transform liquid wastes, including BW, into valuable products for agriculture, while reducing pollution levels in water without the need for technical pre-treatment. The results suggest that a 10% diluted raw BW showed the highest growth rate of algae (0.265 per day) and a nutrient removal efficiency of 99.6% for ammonium and 33.7% for phosphate. With a 50% dilution of BW, the highest COD removal (81%) was observed. In summary, the treatment of BW with Chlorella Vulgaris suggested that the dilution factor of 0.5 followed by microalgae cultivation with a hydraulic retention time of 14 days could offer the highest biomass production for the potential use in agriculture and, in parallel, a way to treat raw BW from source-separation sanitation systems [81].

Pilot plant studies in high-rate algal ponds (HRAP) are also carried out with a capacity of 3000 L using Chlorella vulgaris with a hydraulic retention time of 5 days, as shown in Figure 13. The results demonstrated that the average removal rate of nutrients, such as total organic carbon (TOC), total nitrogen (TN), total phosphorus (TP), was found to be 67%, 76%, and 41%, respectively. For contaminants of emerging concern (CEC), the average removal rates of such compounds were 60% for naproxen (NP), 51% for ibuprofen (IB), 92% for methylparaben (MePB), 76% for 2,4-dihydroxybenzophenone (DH-BP), and 80% for oxybenzone (HM- BP) [82]. The quality of algal cells that makes them an ideal candidate for remediation is their fast growth rate. They can produce oxygen from sunlight and need only minimal nutrients for growth. During phycoremediation, algae modify the quality of water by exerting effects on various biochemical parameters, including: efficient pH correction, change in smell and color, sludge reduction, reduction in biological and chemical oxygen demand, removal of nutrients, and flue gas/carbon mitigation. To summarize, the rising expense of traditional remediation methods to decrease pollution in aquatic and terrestrial environments has prompted the adoption of innovative techniques such as phycoremediation as cost-effective and environmentally acceptable green alternatives.

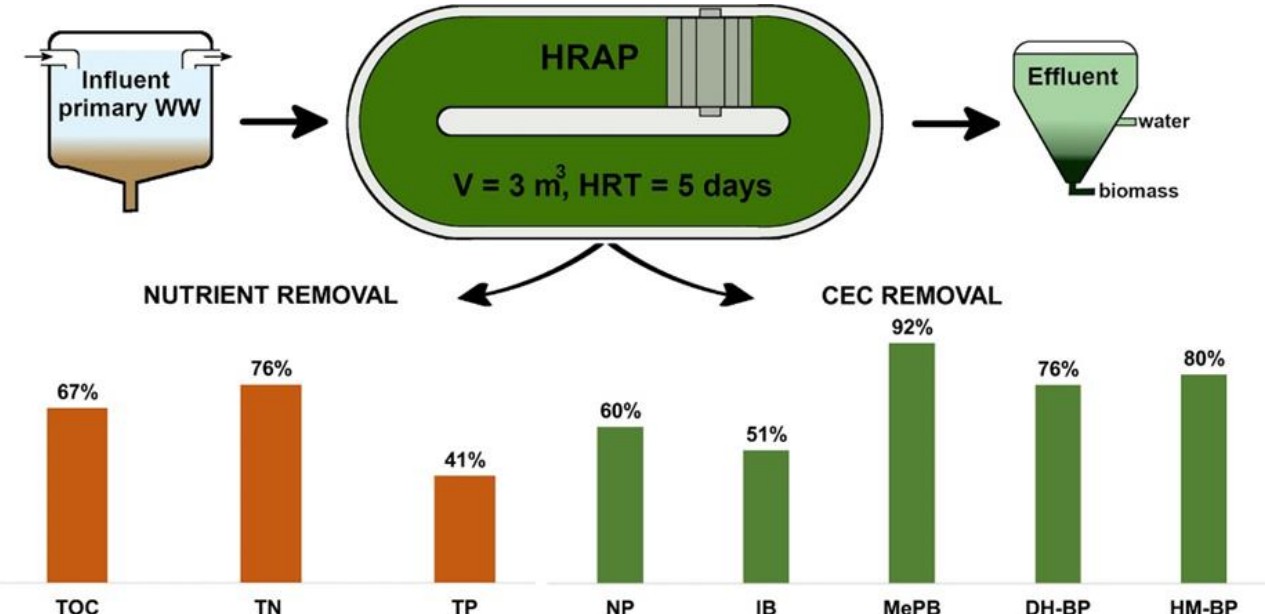

**Figure 13.** Performance of high-rate algal ponds (HARP) for the treatment of municipal wastewater. Adapted from the work of David Škufca et al. [82]. Reused under the Creative Commons license.

## 4. Discussion

In India, SBM or Clean India Mission was a nationwide momentous initiative by the GOI, started in 2014, which has successfully eliminated the paucity of bonafide sanitation facilities in rural areas within five years through the construction of lavatories [83].

This mission also aims to set up an effective solid and liquid waste management system to ensure excellent sanitation facilities in rural regions with sparsely distributed populations [84]. With the effective participation of people, the campaign resulted in the construction of more than 100 million toilets, directly benefiting 500 million people across 630,000 villages by 2019. The success of this mission is now being emulated by other national campaigns in countries such as Nigeria, Indonesia, and Ethiopia [83]. The Bill & Melinda Gates Foundation also released a study that displays a significant improvement in diarrhea prevalence and stunting among children in villages due to the convergence of activities under SBM [85]. The upcoming application steps of SBM must be integrated with novel, cheap, and non-conventional treatment techniques to manage liquid and solid waste, thus ensuring environmental sustainability and a society free from diseases. Since waste management is also one of the primary objectives of SBM, in this work, we have encapsulated the small-scale decentralized processes for handling wastes from lavatories, especially in places with a sparse population. Here, we summarized effective treatment methods specifically to treat BW. These methods ensure the efficient recovery of nutrients as a fertilizer, the removal of BOD and COD, the complete removal of pathogens, the production of biofuel from waste, and the reuse of treated water for a variety of purposes. These recommendations are not limited to the Indian scenario, but they apply to any other locations in the world where the sophisticated construction of centralized sewage treatment facilities is practically impossible. Additionally, we recommend decentralized BW management systems in upcoming urban areas, since they reduce the burden of dealing with large volumes of municipal sewage water (MSW) at centralized treatment facilities. This also reduces the cost of transportation and the intermediate storage of waste or chemicals required for the on-site treatment of wastes [86].

The novel BW treatment techniques in this work are sequenced in such a way that the BW from the community is sent to a sedimentation tank, where it is separated into liquid (supernatant) and sediment (solid residue). The supernatant is subjected to the EC method, whereas the residue is subjected to any of the following methods, depending on the occupation or needs of the particular locality: (i) vermicomposting, (ii) the microwave method, (iii) biogas, or (iv) phycoremediation. The essence of each of the proposed methods is listed below.

EC method: This method is powered by solar energy and does not require any external power, thus reducing treatment costs. The EC treatment delivers non-potable water that can be used for a variety of purposes linked with agriculture and can be recirculated for toilet flushing. The in-depth analysis and fabrication of novel, energy-efficient electrocatalysts can still reduce the energy consumed for the treatment of the polluted water. Similarly, intensive research into ED techniques can enable the facile recovery of nutrients for agriculture.

Vermicomposting: With this method, earthworms convert organic waste into stable fine granular organic manure. This manure helps to enhance the quality of the soil by inhibiting the growth of pathogenic microbes and improving the soil's physiochemical and biological properties. Researchers have reported that the effluent pollutant concentrations resulting from this method have met irrigation water quality standards [87]. The method is straightforward and does not require any advanced machinery, and the methodology can be improved with respect to tank design in the future to provide the optimized treatment of solid residue from BW.

MW reactor: This method reduces the sludge volume by about 70% within 30 to 120 min using microwaves [57,63]. In addition, it completely destroys bacteria and decomposes heavy metals, and the treated material has a high calorific value and nutrient content, making it suitable as a biofuel and soil conditioner. This method can effectively be used for the rapid treatment of sludge in high-altitude areas and emergency settlements. Future works pertaining to the economic analysis of this technique in each characteristic region could potentially enable its implementation for the decentralized treatment of solid residue emerging from the BW.

Biogas-based treatment: The decomposition of organic matter by bacteria in the absence of oxygen produces methane gas and carbon dioxide. This biogas would replace fossil fuels to meet the current energy needs of households, industry, and transportation. Additionally, these fuels emit less $CO_2$ than commercially available fuels, contributing to a more environmentally friendly planet. This method is simple, easy to install, economical, and does not require any sophisticated equipment. These plants can be installed for an individual house or a group of houses, and biogas production will be feasible in regions where temperatures range between 31 °C and 34 °C [88]. The digestate that comes out of the biogas plant can be used as fertilizer in agriculture as a substitute for commercially available chemical fertilizers.

Phycoremediation treatment: This method employs the use of algae to decompose or remove undesirable substances and materials present in the wastewater. This is an effective waste treatment method without the use of harmful chemicals. Algae have been widely used in the bioremediation of pollutants, producing clean water as well as useful biomass that may be used as feedstock for a variety of valuable products. These include food, feed, fertilizer, pharmaceuticals, and more recently, biofuels. Furthermore, algae have the potential to be a carbon sink by removing $CO_2$ during photosynthesis. Since algal sustenance requires high levels of light and temperature, this method is particularly suitable for countries with relatively warm climates. Table 3 presents a comparison of novel diverse methods for the on-site treatment of BW and their corresponding advantages and disadvantages.

**Table 3.** Comparison of diverse novel techniques for the treatment of BW and their corresponding advantages and disadvantages.

| Treatment Techniques | Advantages | Disadvantages |
|---|---|---|
| Electrochemical (EC) | 1. Powered sustainably through solar systems<br>2. Does not require external chemical, as the oxidants (for example: chlorine) are produced in situ within the reactor<br>3. Produces non-potable water suitable for application in agriculture and toilet flushing | 1. Electrode fouling<br>2. Regeneration of membrane for removing the accumulated organic layer leading to discontinuous operation. For example, for study discussed in this work, maximum the EC reactor was operated continuously for was 60 days.<br>3. Can only treat the supernatant phase of blackwater, and the solid sludge is treated by other techniques |
| Vermicomposting | 1. Solid human waste is converted into vermi-cakes, used as manure and fertilizers.<br>2. The pathogen concentration (Total Coliforms (TC) (55.74%), Fecal Coliforms (FC) (53.97%), and Fecal Streptococci (FS) (60.36%)) in the effluents of vermicompost-treated sample are acceptable and meet the irrigation quality standards<br>3. Simple in operation and involves no transportation and treatment cost<br>4. Significant removal of BOD (86.3%), COD (70.2%), and SS (45.7%). | 1. Longer treatment time (~90 days)<br>2. Treatment is possible only in batch mode<br>3. The overflowing liquid from the worm tank should be treated by other eco-friendly techniques like EC systems |

Table 3. *Cont.*

| Treatment Techniques | Advantages | Disadvantages |
|---|---|---|
| Microwave (MW) | 1. Reduces the sludge volume by 70% within 30 to 120 min of treatment<br>2. Complete elimination of pathogens and facile decomposition of heavy metal ions<br>3. The treated dried sludge has high calorific value ($\geq$16 MJ/kg) and nutrient contents (solids: TN $\geq$ 28 mg/g TS and TP $\geq$ 15 mg/g TS)<br>4. Rapid treatment of sludge and effective for implementation in high-altitude areas, deserts, and emergency settlements | 1. Lacks proven global technologies<br>2. Requires massive on-site trails and economic analysis for practical implementation |
| Biogas | 1. Biogas plants are simplistic in operation, easy to install without the requirement of any sophisticated equipment, and highly economical<br>2. Biogas produced from the treatment of sludge from BW has the potential to replace fossil fuels to meet the current energy needs of households, industry, and transportation sector<br>3. Digestate can be used as source of nutrients for agriculture | 1. Treatment is possible in the temperature range of 31–34 °C<br>2. Operation is possible in batch mode |
| Phycoremediation | 1. Algae are used for the bio-degradation and/or bio-transformation of the sludge without the requirement of any external chemicals<br>2. After treatment, produces biomass that can be used as fertilizers and biofuels<br>3. Reduces greenhouse gas ($CO_2$) emissions through photosynthesis, thus acting as a carbon sink<br>4. Exhibits the capacity for the significant removal of the contaminants of emerging concern (CEC); average removal of such compounds: (i) 60% for naproxen (NP), (ii) 51% for ibuprofen (IB), (iii) 92% for methylparaben (MePB), (iv) 76% for 2,4-dihydroxybenzophenone (DH-BP), and (v) 80% for oxybenzone (HM- BP) | 1. Algae growth is highly suitable at tropical temperature<br>2. Requires long hydraulic retention time (>14 days) for exalted production of biomass |

To conclude, for the treatment of BW, the above discussed methods can be combined or used individually, depending on the requirements and limitations of the locality in which these techniques are targeted for their implementation.

**5. Conclusions**

This article discusses effective treatment techniques for treating BW as well as a solution for waste disposal, which is a major concern with respect to pollution. The effective utilization of BW to produce non-potable water for reuse and other value-added products has been of significant research interest in recent years. Decentralized BW management systems would potentially be implemented in diverse areas of the world, including both urban and rural regions, as they reduce the cost of transportation and the burden of managing large volumes of municipal sewage water (MSW) at a centralized treatment

facility. The treatment pattern suggested in this work initially subjects the collected BW to sedimentation and obtains an upper supernatant liquid phase and a bottom solid residue. The polluted liquid phase is treated by EC treatment, and this effectively produces non-potable water with applications for agriculture and lavatory reuse. The bottom solid residue is subjected to either vermicomposting, MW reactor, biogas, phycoremediation. The choice of treatment method for handling the sediment is dictated by the requisites and limitations of that particular locality in which decentralized treatment is implemented. These methods are simple to construct, economical, and do not require sophisticated instrumentation. Furthermore, these methods ensure the satisfactory removal of BOD, COD, and pathogens and the degradation of heavy metals and micro pollutants, subsequently yielding value-added products such as enriched organic fertilizer, biogas, biofuel, and non-potable water. These methods are not specific to any region and can be used across the globe in locations lacking centralized sewage treatment facilities.

**Author Contributions:** P.R.A. provided the conceptualization, supervision, and writing—original draft preparation. D.J.P.K. provided the supervision and writing—original draft preparation. A.D.H.V.O.J. provided the writing—original draft preparation. N.S.P. provided the writing—original draft preparation. J.V. provided the writing—the original draft preparation. A.M. provided the writing—review and editing. All authors have read and agreed to the published version of the manuscript.

**Funding:** This research received no external funding.

**Data Availability Statement:** Not applicable.

**Conflicts of Interest:** The authors declare no conflict of interest.

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
