# Peer review of "Towards Viable Eco-Friendly Local Treatment of Blackwater in Sparsely Populated Regions"

_water, doi:10.3390/w15030542_

Round 1

Reviewer 1 Report

The manuscript provides a review of eco-friendly local wastewater treatment approaches suitable for SMB, India. The topic and content discussed in the paper are aligned with the journal's aim and scope and should be considered for publication. However, there are several issues and missing points that require addressing before the manuscript can be reconsidered. Given that this is a review paper, the expectation is comprehensive recent literature is used to elaborate on discussions. Further, the language of the paper requires enhancement and improvement, to eliminate vague and inappropriate language. for example, L12 'Efficient durability of people’s lifestyle' doesn't make any sense! Follow the below point-by-point comments to revise your manuscript: 

Abstract: you need to justify why EC is your method of choice and back it up with evidence and robust discussion L29-31. 

Keywords: wastewater treatment and sustainable water treatment seems important missing keywords. These should help you to ensure broad readership will be achieved.

Introduction:  The discussions and literature used to back the arguments need to be enhanced in the Intro section. The discussions can be quite basic in some paragraphs (e.g.L37-40). 
L47-50, you need to discuss the human health risk of being exposed to or consuming polluted water, such as the carcinogenic risk of NO2-NO3 (10.1016/j.jclepro.2022.132432, 10.1016/j.jtice.2021.01.030).  
L51-54, you also need to add references concerning nature-based solutions such as ponds and baffled ponds (10.1016/j.ecoleng.2022.106702; 10.1016/j.jwpe.2020.101411). 
L55-59, add references for WWTP (10.1007/978-3-319-99867-1_122; 10.1007/s11270-018-4053-1).
L86-89,  can you provide a link to the Government of India (GOI) Swachh Bharat Mission (SBM), so the reader can better understand SBM without the need of making this paper longer for such discussions? 
Fig. 3 - the census data presented in this figure is not very clear, can you improve this figure, or provide statistics for each state?

2. Prospective of On-Site Treatment Processes: L143-145, you need to mention the benefit and sustainability of your proposed system compared to traditional WWTP that rely on massive energy consumption (10.2166/wst.2020.220; 10.1007/s11356-020-08277-3).

3. Novel Diverse Techniques for the On-Site Treatment of Black Water (BW): This is very good, can you provide a table to summarize the key benefits/advantages of each method as well as their limitations? Also, statistics about these methods' performance would be very useful. 
Fig. 6 may not be very clear for a typical water engineer without chemistry knowledge, can you either enhance the figure or ensure the discussions related to the figure is very clear. 
I think section 3 is comprehensive and well described, statistics and comparative performance analysis between the methods discussed can be really helpful for the readers. 

Concluding remarks:  are mostly well-written, you should put more effort to highlight study significance and new knowledge/ contributions.

Finally, upon revision of the above points, you are required to carefully proofread the manuscript and improve the writing. 

Author Response

Dear Reviewer,

Thanks a lot for your comments and suggestions! We have answered your valuable comments and have attached the file carrying the point-by-point response file for your kind perusal.

With Regards,
Authors.

Reviewer 2 Report

This article reviewed recently reported low-cost purification technologies for recycling wastewater generated by single and community households in sparsely populated areas. Comparison and suggestions were provided to give insight into the future development and application of these methods for effective waste management that can be conveniently integrated into toilets built under the Swachh Bharat Mission (SBM) in the coming years. The structure of this paper could be modified to make it easier to follow. More discussion and comparison could be provided to look out the future development.

Major comments:

1. The author should check the grammar carefully.

2. Section “2. Prospective of On-Site Treatment Processes” could be move to the discussion or conclusion.

3. A comparison table could be provided in the discussion section to list and directly show the advantages and disadvantages of these methods.

Specific comments:

Line 13: change “recycle” to “recycling”.

Line18: change “ensures” to “ensure”.

Line 28: specify “the satisfactory”. Give a number or range to the review’s results.

Line 31: change “to” to “in”, “areas” to “area”.

Line 68-69: delete “number of”; change “attributing” to attributed”.

Line 77: “led to a rapid”.

Line 86: “provide them with a healthier living culture”.

Line 95-95: “such a way”; “cleaning the toilet”; “such a mass”.

Line 111: change “are” to “is”. 

Author Response

(The authors gave the same response as above.)

Round 2

Reviewer 1 Report

The authors have revised the manuscript according to the comments. The work is very informative and provides a comprehensive review of the existing options for eco-friendly blackwater treatment in sparsely populated regions. I think the manuscript is now ready for publication. However, language edits and proofreading will be required.